# Oil Bioremediation in the Marine Environment of Antarctica: A Review and Bibliometric Keyword Cluster Analysis

**DOI:** 10.3390/microorganisms9020419

**Published:** 2021-02-17

**Authors:** Nur Nadhirah Zakaria, Peter Convey, Claudio Gomez-Fuentes, Azham Zulkharnain, Suriana Sabri, Noor Azmi Shaharuddin, Siti Aqlima Ahmad

**Affiliations:** 1Department of Biochemistry, Faculty of Biotechnology and Biomolecular Sciences, Universiti Putra Malaysia, Serdang 43400, Selangor, Malaysia; nadhirahairakaz@gmail.com (N.N.Z.); noorazmi@upm.edu.my (N.A.S.); 2British Antarctic Survey, NERC, High Cross, Madingley Road, Cambridge CB3 0ET, UK; pcon@bas.ac.uk; 3Department of Chemical Engineering, Universidad de Magallanes, Avda, Bulnes 01855, Chile; claudio.gomez@umag.cl; 4Center for Research and Antarctic Environmental Monitoring (CIMAA), Universidad de Magallanes, Avda, Bulnes 01855, Chile; 5Department of Bioscience and Engineering, College of Systems Engineering and Science, Shibaura Institute of Technology, 307 Fukasaku, Minuma-ku, Saitama 337-8570, Japan; azham@shibaura-it.ac.jp; 6Department of Microbiology, Faculty of Biotechnology and Biomolecular Sciences, Universiti Putra Malaysia, Serdang 43400, Selangor, Malaysia; suriana@upm.edu.my; 7National Antarctic Research Centre, B303 Level 3, Block B, IPS Building, Universiti Malaya, Kuala Lumpur 50603, Malaysia

**Keywords:** fuel, biotechnology, pollution, ocean, microorganisms

## Abstract

Bioremediation of hydrocarbons has received much attention in recent decades, particularly relating to fuel and other oils. While of great relevance globally, there has recently been increasing interest in hydrocarbon bioremediation in the marine environments of Antarctica. To provide an objective assessment of the research interest in this field we used VOSviewer software to analyze publication data obtained from the ScienceDirect database covering the period 1970 to the present, but with a primary focus on the years 2000–2020. A bibliometric analysis of the database allowed identification of the co-occurrence of keywords. There was an increasing trend over time for publications relating to oil bioremediation in maritime Antarctica, including both studies on marine bioremediation and of the metabolic pathways of hydrocarbon degradation. Studies of marine anaerobic degradation remain under-represented compared to those of aerobic degradation. Emerging keywords in recent years included bioprospecting, metagenomic, bioindicator, and giving insight into changing research foci, such as increasing attention to microbial diversity. The study of microbial genomes using metagenomic approaches or whole genome studies is increasing rapidly and is likely to drive emerging fields in future, including rapid expansion of bioprospecting in diverse fields of biotechnology.

## 1. Introduction

Antarctic research plays a key role in understanding and predicting future environmental trends, for instance providing a global barometer and record in fields such as pollution and climate change [1]. Human contact with the Antarctic continent has taken place over only the last two centuries, initially focusing on exploration and commercial exploitation, with the emphasis switching to scientific research in the second half of the 20th century, and more latterly incorporating tourism. Antarctica is remote from other continents and access and support of all human activities in this harsh and challenging region requires the support of ships and aircraft. The Southern Ocean surrounding Antarctica is regarded as one of the most remote and unpolluted marine environments on Earth. A key concern for Antarctic marine environments lies in the danger of release of large volumes of petroleum hydrocarbons [2], in particular through shipping accidents, a number of which have already occurred. Studies have also identified the possibility of long-range atmospheric transport of polycyclic aromatic hydrocarbons (PAH) and other pollutants from other southern landmasses [3].

Antarctica is administered under the Antarctic Treaty, a major international treaty which was negotiated in 1959 and came into force in 1961. Initially with 12 original signatory nations (‘Parties’), the Treaty now has 54 national members, 29 of which are full ‘Consultative Parties’ and take part in the consensus decision making processes of the annual Antarctic Treaty Consultative Meetings (ATCM). Over the treaty’s existence, various environmental regulations have been adopted, with the aim of protecting Antarctica’s environment. The applicable legislation today is the Protocol on Environmental Protection to the Antarctic Treaty, which came into force in 1998 (hereafter referred to as the Environmental Protocol). The Environmental Protocol prohibits the transport or release of certain persistent organic pollutants (POPs) and other chemicals to the continent, and provides the regulations controlling practices at research stations, on vessels and in field locations [4,5].

Bibliometrics is a quantitative analysis approach that uses mathematical and statistical tools to measure the inter-relationships and impacts of publications within a given area of research [6]. It can provide a macroscopic overview of large quantities of academic literature and reliably identify influential studies, authors, journals, organizations and countries over time [7]. Bibliometric mapping provides a means to visualize academic output as publication and citation information for parameters of a particular field. It allows the representation of information in a manner that can clarify relationships and lead to new insights and discoveries [8].

There is currently a lack of bibliometric mapping and clustering analysis of research in the field of oil bioremediation in the maritime Antarctic, a region that includes the Antarctic Peninsula and Scotia Arc archipelagoes and the ocean surrounding them. The main objective of the current study was to analyze a dataset constructed from the ‘ScienceDirect’ publication database related to oil pollution and bioremediation in the marine environment of the maritime Antarctic between the years 1970–2020, with a primary focus on the last two decades since 2000.

## 2. Review Methodology

We undertook a bibliometric analysis of keyword co-occurrence. Data mining and database construction was carried out using VOSviewer (Centre for Science and Technology Studies, Leiden University, Leiden, The Netherlands) version 1.6.16 analysis software (https://www.vosviewer.com (accessed on 31 December 2020)) applied to the ScienceDirect publication database https://www.sciencedirect.com (accessed on 31 December 2020)). The VOSviewer software developed by [7] is free for public use and allows the investigation through mining of the title, abstract and key words of papers [9] An initial analysis was carried out identifying all relevant publications since 1970. Subsequently, detailed analysis was limited to publications between the years 2000 and 2020, relating to oil bioremediation and oil pollution in the geographical region of the maritime Antarctic. This includes the western coastal regions of the Antarctic Peninsula and the Scotia Arc archipelagoes of the South Shetland, South Orkney and South Sandwich Islands and the remote Peter I Øya and Bouvetøya. Search terms used in the thematic search included the following combination: ‘Antarctic’ and ‘marine’ or ‘Southern Ocean’ or ‘seawater’ and ‘bioremediation’ and ‘hydrocarbon’ or ‘diesel’ or ‘crude oil’ or ‘fuel’ or ‘oil’. The literature type was defined as ‘All types’ which, according to ScienceDirect database, consists of review articles, research articles, conference abstracts, correspondences, editorials, mini reviews, short communications, data articles and book chapters. All the data files were downloaded on 14 November 2020.

Using VOSviewer analysis software, keyword co-occurrence was assessed based on publications listed in the ScienceDirect database for the years 2000–2020 using the full counting method. Keyword co-occurrence analysis is a method of effectively mapping the strength of association between keywords in contextual data [9]. The minimum number of occurrences for any given keyword was selected to be 5. Of the 8732 keywords identified, only 1236 met this threshold. For each of the 1236 keywords, the total strength of the co-occurrence links with the other keywords were calculated. A total of 218 keywords contributed to the greatest total link strength. Each keyword is displayed on the diagrams (maps) produced as its own node or circle, each of which are joined by line(s) also known as links.

VOSviewer provides three different visualizations of the analysis; network visualization, density visualization, and overlay visualization. This review used network visualization to provide an overview of keyword co-occurrence through total link strength (TLS). A link means a co-occurrence connection between two keywords. Each link has a strength represented by a positive numerical value [10]. The higher the value, the stronger the link, thus the thicker the line illustrated. The total link strength indicates the number of publications in which any two keywords occur [7].

Density visualization was used to illustrate the density of each of the 218 Keywords (items). The color (density) of a given point depends on the number of items in the neighborhood of that point and on their importance. The density visualization helps provide an overview of the general structure of the keyword map and to identify the most important areas [7]. Density visualization was calculated using a complex derivation of Equation (1) [10] where K denotes the Gaussian kernel function and t is a function of time.
K(t) = exp(−t^2^)(1)

Cluster mapping was used through overlay visualization which allowed the keywords to be sorted into clusters. Cluster mapping is a method within the VOSviewer software that divides the keywords into clusters of well-represented subject matters [10] using a timeline-based approach. The location of any given circle in the cluster analysis is determined based on the specific point in time of source publication and its relationship to other circles [7]. Weight of measurement was selected as the co-occurrence of keywords.

## 3. Publications Relating to Oil Pollution Affecting Marine Environment of Antarctica, 1970–2020

Sporadic hydrocarbon pollution events affecting the Antarctic marine environment have been documented. The most notable such event happened on 28 January 1989, when the Argentinian supply vessel *Bahia Paraiso* ran aground in the Bismarck Strait near the coast of Southern Anvers Island west of Antarctic Peninsula near the United States’ Palmer Research Station. This incident involved the largest and most damaging oil spill ever to have occurred in Antarctica [11]. Approximately 600,000 L of fuel, mostly Diesel Fuel Arctic (DFA), were released into the marine environment, and much was washed up along the local coastline, also affecting local penguin and other marine bird colonies and marine mammals that haul out on these shores [11]. In 2001, a Chilean ship contracted by Ecuador, *Patriarche*, ran aground and spilled 1500 L of diesel fuel off the north-west Antarctic Peninsula coast [12]. The most recent marine spill recorded occurred in November 2007 when the MS *Explorer*, an ecotourism cruise vessel, suffered a hull breach after hitting a growler and sank soon afterwards. The vessel was carrying about 190,000 L of marine gas oil (MGO), 24,000 L of lubricant oil and 1200 L of petrol [13].

Many reports of small-scale fuel spills have been made in the vicinity of Antarctic research stations, the majority of which lie near the coastline [12,14,15]. Such terrestrial fuel spills also pose a risk to the marine environment due to the likelihood of fuel runoff into the nearby marine environment. The largest such example took place in the continental Antarctic in 1989, when a valve failure at the United States’ airfield Williams Field on the Ross Ice Shelf resulted in a spill of 260,000 L of kerosene. During the subsequent clean-up, only 100,000 L of fuel were recovered, and the remainder soaked into the snow and ice, eventually calving into the sea [16]. A further significant spill took place at Casey Station on the East Antarctic coastline in 1990 [15]. Recorded spills have generally been smaller at stations in the maritime Antarctic. For instance, an accidental spill of diesel fuel of approximately 1000 L took place at the United Kingdom’s Faraday Research Station, located immediately adjacent to the coast on Galindez Island in the Argentine Islands in March 1992 [17]. Marine sediment communities can act as ecological indicators of hydrocarbon contamination [14,18]

Figure 1 provides a general overview of publication trends relating to oil pollution and remediation in the marine environment of the maritime Antarctic. The earliest identified report mentioning oil spill and making comparison between ocean samples from Antarctica, the Atlantic and the North Sea was published in the first issue of Marine Pollution Bulletin in 1970 [19]. Publication rates remained low through the 1970s before increasing consistently through the 1980s, with a peak at the start of the 1990s coinciding with the *Bahia Paraiso* shipwreck. However, after that, rates were markedly lower throughout the 1990s, with the exception of 1997. Rates started to increase again through the 2000s, though only returning to 1990 levels around 2008, then staying at or above this level through most of the 2010s before a further increase in the most recent three years to 2020. In the latter years around 40 publications per year were identified. Table 1 provides a summary overview of the main subject areas addressed in publications in the successive decades after 1970. Oceanographic studies and assessment of hydrocarbons in seawater in the 1970s [20] were followed by a progressive increase in chemical studies of hydrocarbon constituents and wider geographical coverage of study sites and types, such as in marine ocean sediments [21]. At the turn of the 21^st^ century, studies of hydrocarbon degrading bacterial species started to emerge [22] as well as an increase in studies of the toxicity of hydrocarbons to Antarctic species [23]. Through the 2000s and 2010s, bioremediation studies employing different microorganisms have become a research focus, along with detailed molecular and physiological studies of hydrocarbon-degrading species [24,25]. Most recently, novel technology and approaches have been applied to improve bioremediation, with consideration of genes, microcosm studies, microbial community studies and biomonitoring studies [18,26,27].

## 4. Types of Fuel Included in Research Published 2000–2020

Table 2 details the published studies and hydrocarbon of interest identified in the maritime Antarctic literature in the period 2000–2020. Overall annual fuel consumption by research stations in Antarctica amounts to approximately 90 million liters [23]. As well as studies focusing directly on oils, the presence of n-alkanes and other hydrocarbons such as toluene and ethylbenzene in Admiralty Bay, King George Island, has also been interpreted as evidence of diesel usage being a local, anthropogenic, source of hydrocarbons in Antarctica [3].

Figure 2 illustrates the different types of fuel oil commonly used in Antarctica and referred to in reports published between 2000 and 2020, and in the latter year alone. Antarctic marine gas oil (MGO) is commonly used in Antarctic and sub-Antarctic regions [45]. The Marine Environment Protection Committee (MEPC) has issued a ban on the carriage of heavy grade oils as cargo or fuel in the Antarctic area. This ban includes crude oil, bitumen and tar and came into effect in 2011, regulated by International Maritime Organisation (IMO).

## 5. Studies of Hydrocarbon Degraders

Bacteria have dominated studies of hydrocarbon degradation, taking advantage of the abilities of some to thrive even under extreme and unexpected conditions [48,49]. Bioremediation takes advantage of the abilities of some microorganisms to reduce toxic compounds present in the environment. Temperature, pH and nutrient bioavailability are crucial factors influencing bioremediation, as they enable certain microorganisms to grow rapidly and utilize hydrocarbons as a carbon source. There has been increasing research interest in the bioavailability of hydrocarbons to the microbial community naturally occurring in low temperature environments, including in Antarctic seawaters. Studies of bacterial hydrocarbon degradation can be grouped into two approaches, those focusing on a single bacterial taxon and those using microcosm methodologies to study bacterial consortia or communities made up of multiple taxa, with the latter gaining more attention in recent years [27,50] (Table 3). Studies have increasingly focused on the population dynamics of marine communities [12,46,47] when exposed to hydrocarbons, moving away from studies requiring the cultivation of specific taxa with biodegradation ability.

## 6. Metabolic Mechanisms of Bacterial Biodegradation of Hydrocarbons

The majority of oil hydrocarbons that enter the marine environment will eventually undergo biodegradation mediated by indigenous microbial communities [52]. Species capable of oil biodegradation are usually present in small numbers in aquatic ecosystems but can rapidly dominate a drastically altered microbial community when exposed to oil pollution. These microbes have high enzymatic capacity enabling degradation of the complex hydrocarbons present in the oil. The genes involved in producing hydrocarbon-degrading enzymes may be located on both plasmid and chromosomal DNA [53].

When a microbial cell encounters oil, the response follows three general stages (Figure 3). The first stage initiates with the physical contact between the oil substrate and surface of the cell. Short-chain, water-soluble alkanes are directly taken into the cell [54]. Medium- to long-chain alkanes and high molecular weight substrates, which are less soluble, are taken in when the microorganisms adhere to hydrocarbon droplets using mechanisms that incorporate surface-active compounds such as biosurfactants, bioemulsifiers, and extra-cellular polysaccharides [55]. These surface-active molecules are produced by the bacteria and act as emulsifying agents that produce micro-droplets of hydrocarbons, enhancing their water solubility and decreasing surface tension, thereby facilitating the translocation of these less soluble substrates across the cell membrane [56].

The second stage involves the transportation of the substrate across the bacterial cell membrane cell into the intracellular space. This can be achieved by passive diffusion, passive facilitated transport and active transport. Certain genes are involved in the uptake of alkanes, such as the alkL gene in *Pseudomonas putida* [57]. Outer membrane proteins of Gram-negative bacteria include transporter proteins that participate in the transportation of long-chain hydrocarbons in many bacteria, such as the long-chain fatty acid transporter protein, fatty acid degradation L (FadL) [58] and outer membrane protein W (OmpW) [59].

The third and final stage is the enzymatic breakdown of the oil components within the cells. Some microbes growing on oil substrates can accumulate lipophilic compounds including unaltered oil substrates and lipids [60] as a survival strategy, through storage as an energy reserve. Furthermore, substrates that support microbial growth are metabolised by enzymes that can act on multiple substrates due to “relaxed substrate specificities’ [61].

### 6.1. Aerobic Biodegradation

A general overview of aerobic catabolism begins with the initiation of terminal and subterminal oxidation of the alkane chain by the enzyme monooxygenase, which results in a primary or secondary alcohol. Further terminal oxidation converts an aldehyde into a fatty acid and eventually into water, carbon dioxide and biomass. Subterminal oxidation also results in fatty acid conversion from ketones and esters [62]. The general process of aerobic bacterial biodegradation is shown in Figure 4.

In aerobic conditions, oxidation occurs through the action of monooxygenase and dioxygenase enzymes. The availability of oxygen as an electron acceptor gives a metabolic advantage, speeding up hydrocarbon catabolism [63]. Saturated alkanes such as aldehydes are oxidized into fatty acids which, in turn, go through β-oxidation to form acetyl-CoA, which can be incorporated in the citric acid cycle together with the production of electrons in the electron transport chain. This sequence is repeated, progressively shortening the hydrocarbon substrate molecule until it is fully oxidized into CO_2_ [63]. In a di-terminal pathway where ω-oxidation occurs at both ends of an alkane molecule, fatty acids go through ω-hydroxylation, being further converted into a dicarboxylic acid which is then subjected to β-oxidation [64]. There are various oxidation pathways available for aerobic biodegradation such as terminal oxidation, sub-terminal oxidation, β-oxidation and ω-oxidation [64]. An essential group of enzymes in the hydrocarbon catabolism process is the alkane hydroxylases. Three categories of alkane-degrading enzyme systems have been proposed [65]: C1–C4 (oxidized by methane-monooxygenase-like enzymes), C5–C16 (oxidized by integral membrane non-hemeiron or cytochrome P450 enzymes), and C17 and above (oxidized by essentially unknown enzyme systems). Microorganisms that can degrade alkanes may contain multiple alkane hydroxylases, thereby being able to utilize multiple substrates.

Low molecular weight (LMW) PAHS such as anthracene, naphthalene, and phenanthrene are commonly present in the environment and have been designated as prototypic PAHs that serve as signature compounds to detect PAH contamination [66]. Compared to high molecular weight (HMW) PAHs, LMW PAHs are more soluble in water making them more easily biodegradable. The biodegradation of PAHS is favored in the presence of oxygen. During aerobic catabolism of aromatic compounds, apart from being a final electron acceptor, oxygen becomes a co-substrate for the hydroxylation and the oxygenolytic cleavage of the aromatic ring. This is initiated by a hydroxylation of the aromatic ring, facilitated by a dioxygenase enzyme, resulting into a cis-dihydrodiol, which is then converted to a diol intermediate by a dehydrogenase reaction. A multicomponent dioxygenase, generally including reductase, ferredoxin, and oxygenase subunits, will then create more ring cleavages in the diol intermediates through either ortho-cleavage or meta-cleavage pathways, resulting in more intermediates such as catechol. The intermediate compounds are ultimately converted to TCA cycle intermediates [67]. Aromatic hydrocarbons such as benzene, xylene, toluene, and naphthalene are aerobically degraded in an initial step to form catechol or a structurally related compound. This is then degraded and introduced into the citric acid cycle where it can be completely degraded into CO_2_ [68]. A further pathway used by bacteria to degrade PAHs is the cytochrome P450-mediated pathway, which results in the production of trans-dihydrodiols [69].

### 6.2. Anaerobic Biodegradation

Aliphatic and aromatic compounds can undergo biodegradation under both aerobic and anaerobic conditions [70]. Although aerobic catabolism generally occurs more rapidly than anaerobic, the latter can be equally important to the bioremediation process because certain environmental conditions are characterized by limited or no oxygen availability, such as in mangroves and sludge digesters [71] and marine sediments [72]. Some marine sediments in Antarctica contain traces of hydrocarbons [11,18] but there is a paucity of data about anaerobes in hydrocarbon-contaminated marine sediments in Antarctica. Biodegradation under anaerobic conditions uses terminal electron acceptors as seen in sulphate-reducing bacteria [70]. As oxygen is depleted, redox reactions proceed with the use of alternative terminal electron acceptors in the following order: O_2_ > NO_3_^−^ > Fe^3+^ > Mn^4+^ > SO_4_^2−^ > CO_2_.

Anaerobic biodegradation of alkanes involves an initial step described as activation, which takes place through one of three pathways, fumarate addition, carboxylation with inorganic carbon, or other possible but unclarified alternative mechanisms [73]. Fumarate is added to the alkane chain to yield a succinate metabolite via a radical enzyme. The metabolite then undergoes a series of skeleton rearrangements to form (2-methylalkyl)malonyl-CoA, which can be decarboxylated to 4-methylalkanoyl-CoA. The resulting fatty acid then undergoes β-oxidation to yield intermediates such as propionyl-CoA and acetyl-CoA. Fumarate is then regenerated from propionyl-CoA or alternatively from acetyl-CoA [74]. Acetyl-CoA is finally oxidized to carbon dioxide. This mechanism is present in most sulphate- and nitrate-reducing bacteria as well as in methanogenic alkane-degrading enrichments [75]. The fumarate addition is an exogenic reaction where ATP hydrolysis is not required. Alkane activation through carboxylation was proposed by So et al. [76], and involves alkanes being metabolized to form fatty acids, which contain a carboxyl group, from an inorganic compound (H_13_CO_3_^−^). However, this activation mechanism remains an assumed possibility as opposed to a confirmed mechanism of fumarate addition due to the unfeasibility of alkane carboxylation under physiological conditions [77].

Other proposed mechanisms of alkane degradation also remain hypothetical. One involves the ability to produce oxygen via chlorate respiration for subsequent alkane metabolism [78]. This is also supported by a suggestion of intracellular oxygen production through nitrate and nitrite electron acceptors [79]. Alternatively, a metagenomic study has suggested hydroxylation of oil alkanes, through the presence of genes encoding oxidation pathways and dehydrogenases as in aerobic degradation [63]. The study also revealed the absence of genes encoding for fumarate or carboxylation pathways, further strengthening the possibility of hydroxylation in aerobic degradation [80].

A microcosm study [81] reported better degradation of 19 and 21C aliphatic chains under anoxic than oxic conditions. In the same study, heavy aliphatic hydrocarbons (>28C) were completely degraded in oxic microcosms but not in anoxic. For anaerobic sulphate- and nitrate-reducing bacteria, short length alkane chains appeared more recalcitrant than did medium to longer length alkanes, accumulating more in the anaerobic environment due to lack of evaporation and, in turn, having toxic effects on the microorganisms and impairing biodegradation. Branched alkanes such as pristane and phytane were more efficiently degraded by sulphate-reducing bacteria than were straight chain alkanes [82].

During anaerobic metabolism, aromatic compounds are generally converted into benzoyl-CoA, which is then targeted by benzoyl-CoA reductase [83]. Depending on the environmental conditions, different terminal electron acceptors can be used, such as nitrate, sulphate and Fe (III) and, generally, the degradation pathways converge to benzoyl-CoA [63]. The Environmental Protection Agency (EPA) lists 16 hazardous hydrocarbons, collectively termed the EPA16 [84]. Amongst these, naphthalene was degraded more effectively than others in oxic microcosms [85], although both phenanthrene and naphthalene were also efficiently degraded under anoxic conditions [86]. In anoxic microcosms, most but not all the EPA16 compounds were degraded, with benz(a)anthracene and chrysene being particularly effectively degraded. A higher proportion of dibenzothiophene was degraded in anoxic than oxic microcosms [81].

Recent reports of anaerobic biodegradation of PAHs have involved compounds such as naphthalene, anthracene, phenanthrene, fluorene, fluoranthene, and acenaphthene [67]. Anoxic conditions dominate many natural and contaminated bodies such as marine and lacustrine sediments and waterlogged soils. Anaerobes can thrive in such habitats, in which PAH biodegradation can become a crucial process. However, developing a generalized overview of the metabolic pathways of PAH degradation has proved difficult due to the complexities and diverse metabolic pathways utilized by anaerobes.

The aromatic ring is attacked primarily through reduction reactions. One pathway includes the addition of fumarate or other carbonic groups to a carbon atom of the hydrocarbon. For example, the addition can be mediated by a benzylsuccinate synthase enzyme, resulting in an unstable radical intermediate. A hydrogen atom from a methyl group becomes a radical, generating the first transition state with the highest energy level and an enzyme-bound radical intermediate. This attacks the unsaturated bond of a fumarate molecule, thus forming a benzylsuccinate. Next, a transferase reaction takes place where CoA is added, followed by additional oxidation reactions finally resulting in benzyl-CoA. This is further oxidized to aliphatic compounds that are subsequently attacked by hydrolytic and oxidative reactions. Similar degradative pathways have been reported for toluene, benzene, xylene and cresol isomers in sulphate-reducing bacteria [87]. Other anaerobic pathways include oxygen-independent hydroxylation [88], carboxylation of unsubstituted carbon atoms [89] and hydration of the double and triple bonds of alkenes and alkynes [90].

Biodegradation of petroleum hydrocarbons in the marine environment is a complex process influenced by external factors including water turbulence, which influences the dispersal of the oil, and salinity. The structure and diversity of the autochthonous microbial community most likely defines their metabolic potential for oil-degradation [91]. In deep-sea environments, hydrostatic pressure plays a synergistic role with low temperature, slowing microbial growth, and oil degradation [92]. The potential for aromatic biodegradation also decreases at high hydrostatic pressure (>15 MPa) [92]. Microbial community composition is also influenced strongly by temperature [93]. Thus, it is critical to understand the relationships between the metabolic potential of hydrocarbon-degrading microbial groups and dynamics of the autochthonous community in the context of the environmental factors at play. The challenge remains to definitively link the structure and function of hydrocarbon-degrading microbial communities in order to improve predictive models of biodegradation [94]. Recognizing the caveats discussed above, Figure 5 illustrates a general pathway for anaerobic biodegradation.

## 7. Analysis of Co-Occurrence of Keywords in Publications between 2000 and 2020

Keywords are nouns or phrases that shed light on the core content of a publication [95]. The number of times an article is cited in other publications is used in a number of indices used to assess its scientific impact. Of the 1236 keyword items identified, 218 keyword items were sorted into 24 clusters as visualized in Figure 6, with the remainder being isolated keywords. This visualization included 837 links with a total link strength of 3753.

In this type of network, the size of the circles reflects how often they occur, and the relative distance (distant or close) indicates the co-occurrence in a given quantity of publications [96]. Also, the bigger the circle, the bigger weight of measurement. The thicker the line co-joining two circles, was often associated with keywords located close to each other in Figure 6 occurring together in the same published articles, while terms that were located distant from each other rarely or never occurred together. Of the 24 clusters identified by the software, 9 colors are indicated in Figure 6b. Colors are used for circles to suggest sets of publications sharing a similar topic. The terms ‘biodegradation’ (green cluster) and ‘bioremediation’ (yellow cluster) had the strongest total links after the obvious keyword ‘Antarctica’ (brown cluster). In general, terms in the center of the map co-occurred with many other terms and were therefore related to multiple topics. In contrast, terms at the edges of the map tend to co-occur only with a small number of other terms; thus, the further the term from the center, the smaller the research focus. Terms at the edges therefore relate to (currently) relatively isolated fields such as ‘west Antarctic Peninsula’, ‘16s RRNA’, ‘molecular markers’ and ‘King George Island’. These isolated terms make up the remaining of the 24 clusters, colored in grey.

In the red cluster, the term ‘psychrophiles’ appeared stronger than ‘psychrotroph’, which could indicate a growing interest in deep sea bacteria that are classified mostly as psychrophiles. Keywords sitting close together, such as ‘metagenomics’ and ‘microbial diversity’ in the red cluster, signal a relationship between them. The term ‘biostimulation’ in the yellow cluster was also relatively strong, implying that many publications recognized its importance in methods of ocean bioremediation over those including natural attenuation and bioaugmentation. However, under the current provisions of the Environmental Protocol, no non-native species can be introduced to the Antarctic environment. The term ‘biostimulation’ can be related to laboratory studies with no natural environmental component and to studies including other global regions remote from Antarctica. The orange cluster includes the terms ’Amundsen sea polynya’ and ‘Macquarie Island’, ‘field experiment’ and ‘oil’, referring to hydrocarbon studies in those specific areas of Antarctica. Note that these terms are located distant to each other even though they are classified in the same cluster.

The green cluster draws together studies of the Southern Ocean, with terms such as ‘marine pollution’, ‘diatoms’, and ‘marine species’. Anthropogenic activities in Antarctica are linked in the brown cluster which includes terms like ‘shipping’, ‘fisheries’ and ‘tourism’, the strongest word being ‘Antarctica’. Next, the aquamarine cluster identified largely by the term ‘biosurfactant’ signifies a growing interest in biosurfactants from cold-adapted microorganisms, with a further link connecting to ‘bioprospecting’ (grey cluster’). The terms ‘algae’ and ‘photobioreactors’ are also in the aquamarine cluster but situated more closely to the green cluster relating to studies of the Southern Ocean. Terms like ‘penguins’, ‘seals’ and ‘Antarctic fish’ are clustered together in pink with links to the cluster in purple. Here, the term ‘toxicity’ appears strongly relating to hydrocarbon toxicity, with many of the published studies adressing the effects of hydrocarbons on Antarctic species. Terms such as ‘sediment’, ‘ocean’ and ‘diesel oil’ appear in the blue cluster, consistent with marine studies in Antarctica including both seawater and marine sediments.

Figure 7 illustrates a density visualization of the occurrence of keywords. It is interesting to note that certain regions and locations on the Antarctic continent stand out, such as Macquarie Island, Collins Bay, West Antarctic Peninsula, and the Ross Sea. The keywords ‘bioremediation’, ‘toxicity’, ‘biodegradation’, ‘sediments’ appears in high density parts of the visualization, suggesting a research focus on the subject of oil pollution in the Antarctic marine environment. Other high density elements relate to the subjects of microbial diversity and enzyme biotechnology. ‘Metagenomics’ refers to the study diversity in the natural environment, an element of Antarctic research that has developed since the term first appeared in 1998 [97].

## 8. Bibliometric Analysis: Focus and Research Direction

Analysis of co-occurrence of keywords enabled the detection of emerging themes in studies of oil bioremediation in maritime Antarctica. Figure 8a illustrates the co-occurring links between terms (keywords) in the entire dataset using an overlay visualization. The many keywords identified in yellow and orange colors in Figure 8a emphasize the upsurge in publications since 2010. This was preceded by increasing interest in the study of microbial species and hydrocarbon degradation through the 2000s, indicated by blue and green colors.

Cluster analysis aims to detect the natural division of networks into groups (clusters) based on similarity and to minimize intercluster similarity. Clusters consist of items (i.e., the 218 keywords) shortlisted in Section 7. VOSviewer software generated 24 clusters (Figure 8a) but only three clusters are discussed here due to the importance of certain keywords appearing in them. Cluster 1 includes 31 items (Figure 8b). Keywords including ‘biotechnology’, ‘enzymes’, ‘microbial diversity’, ‘metagenomic’, ‘metabolomic’, ‘metaproteomic’ and ‘cold adaptation’ imply strong interest in the last decade in the genes, proteins and enzymes from cold-adapted microorganisms in relation to hydrocarbons.

Cluster 24 (Figure 8c) includes three keywords: ‘Casey station’, ‘Special Antarctic Blend’ (diesel) and ‘hydrocarbon’. This cluster was distinct from other clusters, suggesting a specific set of publications pertaining to Special Antarctic Blend (SAB) diesel at Casey Station, operated by Australia on the East Antarctic coastline [37,98]. Cluster 5 (Figure 8d) included the keywords ‘seawater’, ‘hydrocarbon’, ‘oil spills’, ‘biomonitoring’, ‘crude oil’, ‘degradation’, ‘sediments’, ‘toxicity’ and ‘toxic metals’. The emergence of the latter two keywords illustrates interest in the topic of oil spills and hydrocarbon-associated pollution.

Five general keywords were heavily used: ‘Antarctica’, ‘Southern Ocean’, ‘hydrocarbons’, ‘pollution’ and ‘Antarctic’. Table 4 lists the keywords occurring on more than 5 occasions generated by VOSviewer software. The top eight strongest keywords were ‘Antarctica’, ‘Southern Ocean’, ‘Antarctic’, ‘hydrocarbons’, ‘pollution’, ‘bioremediation’, ‘biodegradation’ and ‘bacteria’.

Petroleum, in the context of petroleum fuel, generated the highest co-occurrences of keywords, followed by crude oil, diesel and, lastly, lubricant oil. The keyword ‘diesel’ appeared in many term forms, indicating its importance as a subject in Antarctic research.

Historically, studies have focused on the types of fuel oils being used and carried in the Southern Ocean, such as McDonald et al. [99] and Payne et al. [45]. Under the terms of the Environmental Protocol, oil exploration or extraction is not permitted in the Antarctic Treaty area. The heaviest residual fuel oil that may be carried in Antarctic Treaty waters is an intermediate grade (IFO 180) that is also regulated by the International Maritime Organisation (IMO) who have established high-level protection specifically for Antarctic waters [100]. These intermediate residual fuel oils, along with light diesel and gas oils, are commonly carried as bunker fuels in the Antarctic and sub-Antarctic [45].

## 9. Emerging Trends in Genomic Studies of Antarctic Marine Microorganisms: Prospects and Challenges

Studies of microbial genomes have been revolutionized in concert with the rapid development and reducing costs of DNA sequencing technologies. However, most of these technologies produce short-read data, defined as reads of a few hundred bases in length at most [101]. Such short-read data have specific and limited uses in fields such as strain typing, outbreak tracing and pan-genome surveys [101]. However, the utility of these studies is also strongly dependent upon the availability of accurate and complete genomic sequences, which have been until recently expensive to produce. Genome assembly reconstructs a genome from many shorter sequencing reads [102,103]. The challenge lies in the repeat of two types of assembly algorithm, global and local. A global repeat is the prokaryotic rDNA operon, which typically has a length of less than 7 kilobase pairs (kbp) [104]. In contrast, a local repeat is a simple sequence unit, sometimes only a few base pairs in length, that can be repeated many times in tandem. The challenge these repeats pose for genome assembly can be addressed by increasing read length, thereby resolving more repeats [105].

Many microbial (including viral) studies require the development of cultures and inherently rely on good laboratory practices. Nevertheless, the great majority (>99%) of microbial diversity has yet to be successfully cultured [106]. The development of metagenomics offers considerable advantages over traditional methods, in particular not depending on the ability to develop cultures. Metagenomics is a next-generation sequencing technology that allows the characterization of microbial (and other) communities and their structure on a vast scale. The metagenomic approach is the study of the genetic material present in its entirety in an environmental sample. It is often used in microbiome studies, with the objective of characterizing the composition of microbial communities [107]. Comparative metagenomics makes it possible to identify microbial groups with particular functional features characteristic of given environments. However, one important caveat that is particularly pertinent to studies of communities of extreme environments, such as those of the Antarctic, is that metagenomics identifies only the presence of a particular DNA sequence (itself limited by the coverage of available sequence databases), and not whether that DNA represents a viable organism or propagule. Identifiable DNA sequences are known to survive intact for very long periods either under frozen or cold desert conditions in the Antarctic, even being recognized as a long-term potential source of human-assisted contamination of the overall Antarctic microbiome [108].

Modern-day metagenomic approaches have developed in parallel with metaproteomics, metatranscriptomics, metabolomics, fluxomics, and metaphenomics. Metagenomic studies utilize different genomic approaches to characterize the microbial communities in environmental samples to reveal the genomes of uncultured microbes as well as diversity of taxonomically and phylogenetically relevant genes, catabolic genes and whole operons [109]. Metagenomics can offer a powerful lens for viewing the diversity of the microbial world in regions such as the Southern Ocean, chronically poorly characterized in terms of its microbiology. Metagenomics enables systematic study of marine microbial diversity, and the addressing of taxonomic as well as functional research questions relating to the marine microbiota [34].

Metagenomic analysis utilizes a hypervariable region of the 16s rRNA gene that is used as a phylogenetic marker. The 16s rRNA gene is highly conserved and ubiquitous, a key advantage in most RNA- and DNA-related studies. Early molecular work was conducted by Norman R. Pace. He, along with colleagues, attempted to explore the diversity of ribosomal RNA sequences using PCR [110] leading to the breakthrough proposal of cloning DNA directly from environmental samples [111,112]. Comparisons between metagenomes can be made in terms of GC-content or genome size to document levels of sequence composition, taxonomic diversity or functional complement. To compare population structure and phylogenetic diversity, phylogenetic marker genes such as 16S rDNA can be used. In low diversity communities, genomic reconstruction from a metagenomic dataset can be carried out to compare population structure and phylogenetic diversity [113]. Recent advances in computational statistics have also made functional comparisons between metagenomes possible by comparing sequences against reference databases using Clusters of Orthologous Group (COG) or Kyoto Encyclopedia of Genes and Genomes (KEGG) [114]. Such an approach emphasizes the functional complement of the entire community as a whole, rather than just individual taxonomic groups [113], providing researchers with a tool to investigate the effects of habitat and environmental conditions on community structure and function [115].

## 10. Application of Metagenomics in Marine Bioremediation

One application of metagenomics has been in the study of the dynamic shifts that occur in the marine microbial community when exposed to hydrocarbons [116]. Metagenomic studies can clarify how microorganisms are distributed within a contaminated environment, revealing the key-players and their potentially synergistic roles within a contaminated site [117]. Vázquez et al. [14] documented patterns of microbial community change in response to a hydrocarbon spill, in turn also identifying indicators of the concentration of hydrocarbons present. Metagenomic approaches have led to the identification of novel biodegradation pathways and, to an extent, the efficacy of bioremediation processes [27]. These approaches have also been applied to determining both the microbes and their novel gene families involved in the bioremediation of many xenobiotics [116,118,119].

Many studies have constructed metagenomic libraries for use in screening and identification of genes involved in bioremediation [109]. Such studies provide the initial stage in the identification and modification of microbes with valuable roles in enzyme, lipid and protein production, including biosurfactants, thereby enhancing bioremediation processes. Metaproteomics have been applied to gain insight into the key proteins involved in the physiological responses of microorganisms exposed to various pollutants, again paving the way for the discovery of new metabolic pathways [120]. Metatransciptomics is a tool used to describe microbial mRNA transcriptional profiles to gain understanding into the functions of environmental microbial communities [121]. Metabolomics is another tool used to analyze changes in cellular metabolites present within microbial cells during metabolism of pollutants [122]. At the forefront of current molecular studies is the metaphenome, which is the product of expressed functions that are encoded in metagenomes in the available environment. As suggested by Jansson et al. [123], metaphenomics encompasses the entire ‘omics’ field including metagenomics. This means that, with future advances the ‘omics’ fields generally, it will become progressively easier to identify and track metabolites and molecules among the organisms, thus enabling better integration across all the fields involved [123].

As understanding of the microbial world and the key roles microbes play in ecosystems increases, researchers need to rely on accurate data to understand and improve strategies of identifying and monitoring the impacts of anthropogenic activities. Metagenomics can aid in improving assessments of the recovery of polluted sites and understanding of how microbial communities respond to pollutants in their surrounding environment. Metagenomic data also serve to increase the probability of bioaugmentation or biostimulation trials succeeding [124] by enabling better understanding of the geochemistry of the contaminated site along with that of the microbial communities that are key in providing the required biodegradation pathways.

## 11. Whole Genome Studies

Whole genome analyses can reveal unique insights into functionalities such as degradation abilities, siderophore production, micronutrient scavenging, stress tolerance, protein folding, and structural adaptation of enzymes and lipids [125]. Having detailed knowledge of the abilities of a microorganism of interest could lay the foundation for developing biotechnological applications to harvest their potential, including the use of genetic engineering technologies for environmental and commercial usage. Whole genome analysis of the bacteria *Oleispira antarctica* has identified it as being a good candidate for the development of new approaches to the mitigation of oil spill impacts in polar areas [36]. Cold adaptation mechanisms and the regulatory systems involved can also be thoroughly studied through genomic analyses [126]. Metabolic features and genes related to resistance, motility, chemotaxis, nitrogen metabolism, aromatic compound metabolism, and stress responses were amongst the subjects addressed in a recent study of an Antarctic marine bacteria [127]. Table 5 summarizes a range of studies published in the period 2000–2020, demonstrating the scale of application of modern genomic studies in diverse fields of biotechnology.

## 12. Conclusions

This review set out to evaluate research publication trends relating to oil pollution in the Antarctic marine environment over the years 2000 to 2020 based on the ScienceDirect database. Based on keyword analyses, the main research areas identified were in the domains of (a) biodegradation of (b) diesel and crude oil, and that the majority of suitable microbes are (c) bacteria. Emerging keywords in more recent years suggest research trending towards the study of microbial diversity and metagenomics. The application of bibliometrics can support the development of future research in several ways. It helps researchers to identify growing interest in various fields through the introduction of new topics as indicated by keywords and may encourage the emergence of new collaborations and development of new research fields, both pure and applied. In terms of publication trends, studies relating to hydrocarbon pollution and degradation in the Antarctic marine environment show an accelerating upward trend, highlighting the urgency of developing more effective spill and bioremediation responses to better protect the unique and fragile ecosystems of Antarctica, a key founding principle of the Antarctic Treaty.

## Figures and Tables

**Figure 1 microorganisms-09-00419-f001:**
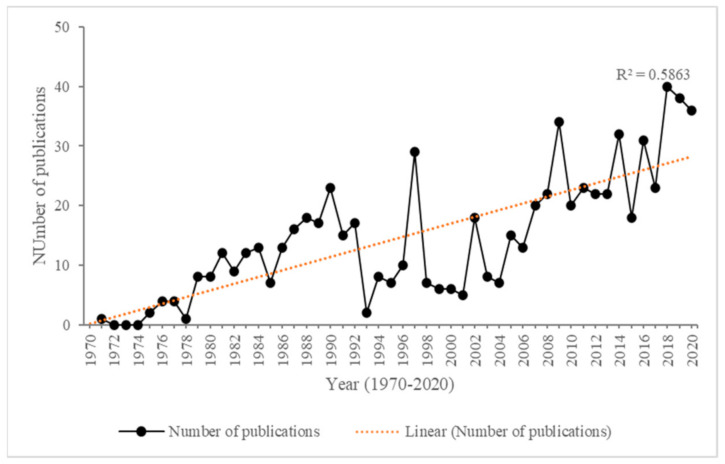
Annual numbers of publications relating to oil pollution and bioremediation in the marine environment of Antarctica in the ScienceDirect publication database between 1970 and 2020.

**Figure 2 microorganisms-09-00419-f002:**
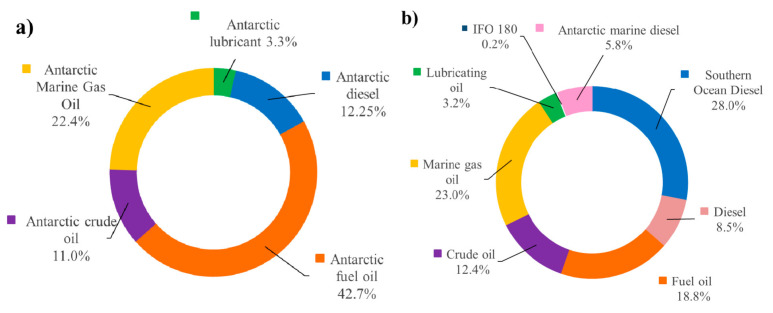
Percentage (%) of different fuel types referred to in the published literature: (**a**) 2020; (**b**) 2000–2020.

**Figure 3 microorganisms-09-00419-f003:**
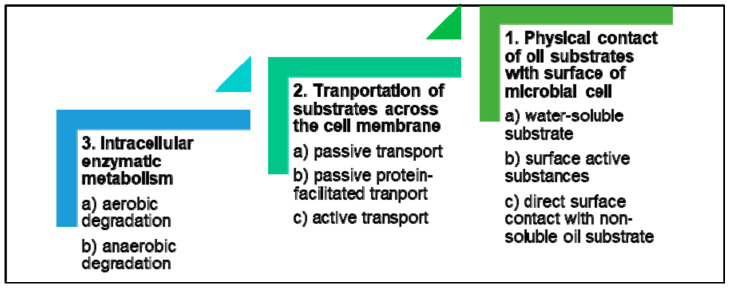
General response stages followed by microbial cells exposed to oil substrates.

**Figure 4 microorganisms-09-00419-f004:**
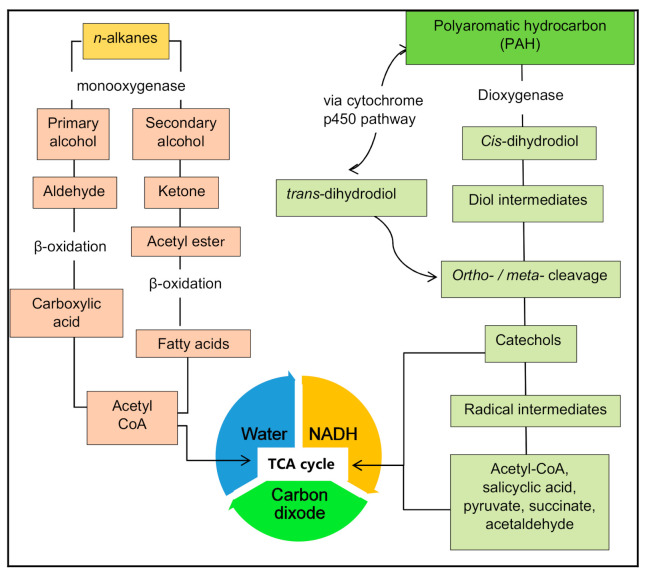
General pathways of aerobic biodegradation of hydrocarbons.

**Figure 5 microorganisms-09-00419-f005:**
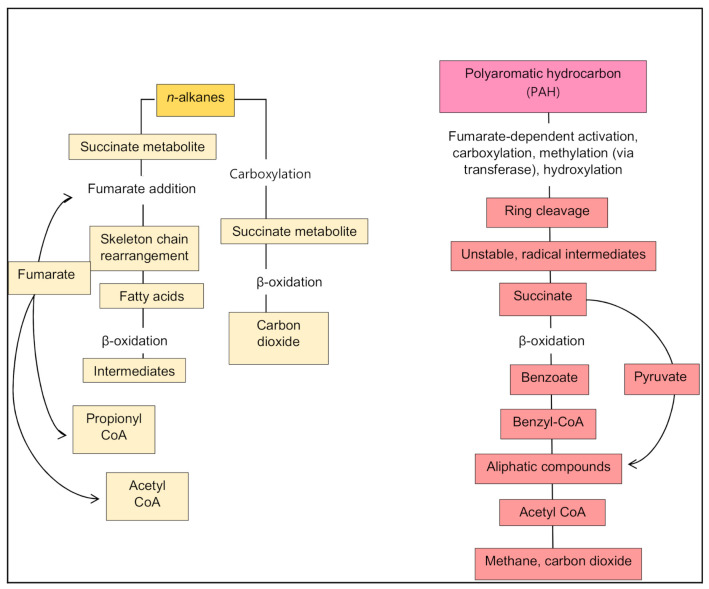
General pathways of anaerobic biodegradation of hydrocarbons.

**Figure 6 microorganisms-09-00419-f006:**
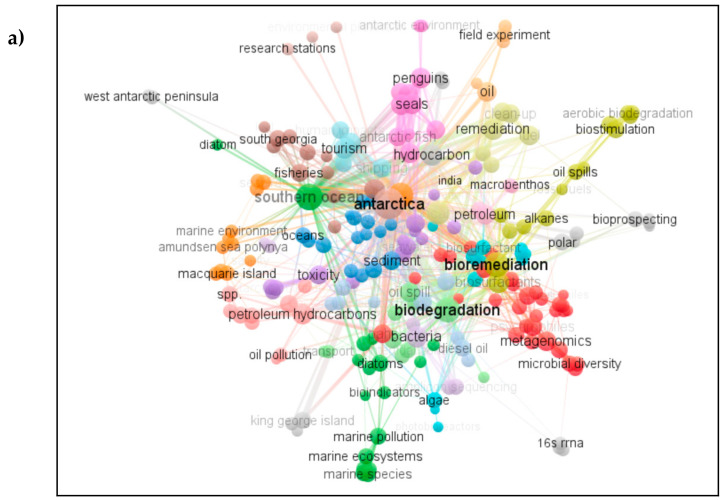
(**a**) Network visualization according to total link strength attribute of the keywords identified; (**b**) color keys indicating subject area of cluster.

**Figure 7 microorganisms-09-00419-f007:**
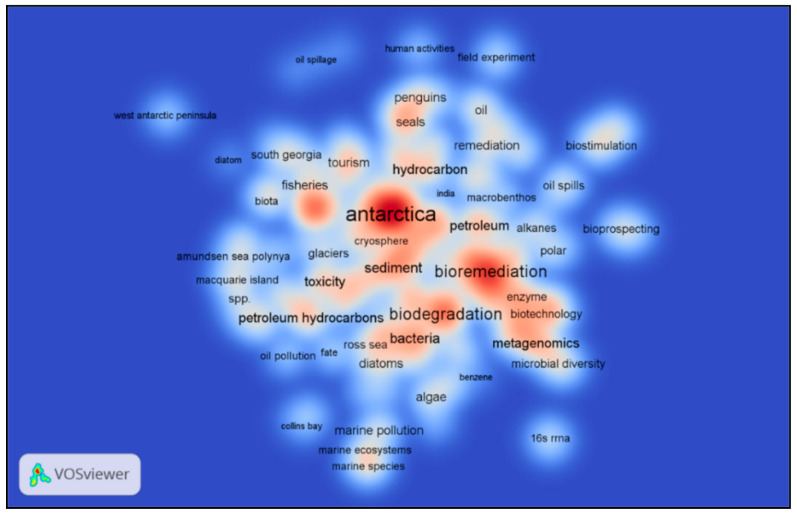
Density visualization for keywords identified in publications in the ScienceDirect database in the years 2000–2020, relating to the search terms ‘Antarctic’, ‘Marine’, ‘Oil’, ‘Fuel’ and ‘Ocean’. Progressively increasing density is indicated by deeper red coloration.

**Figure 8 microorganisms-09-00419-f008:**
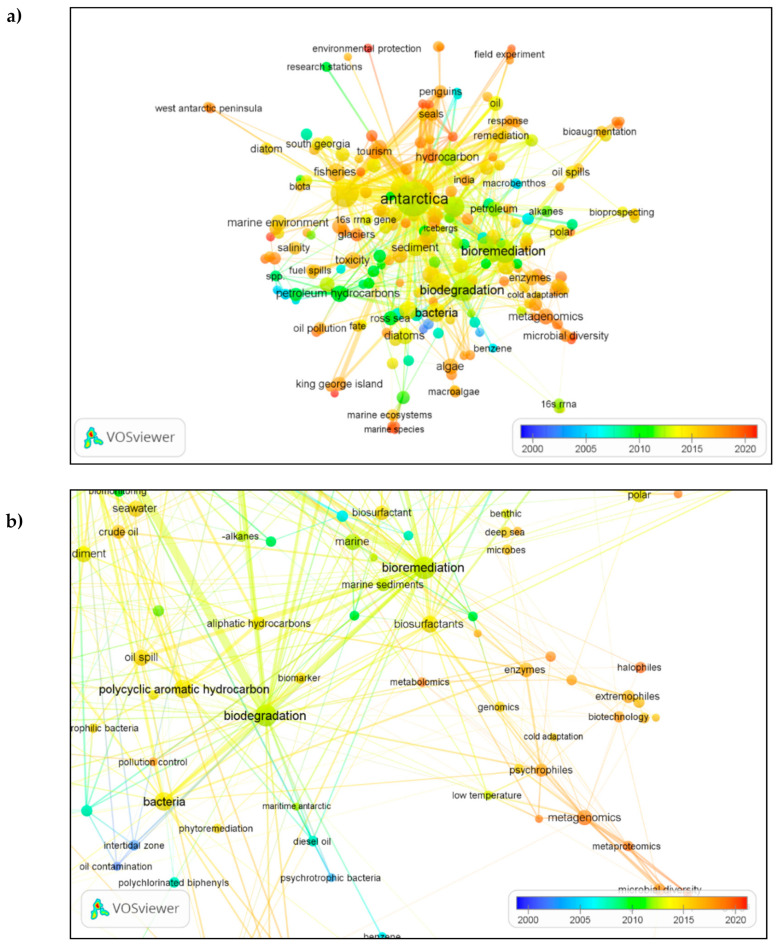
Overlay visualization: (**a**) keywords based on their occurrences and average publication per year scores; (**b**) Cluster 1 with 31 keywords; (**c**) Cluster 24 with three items; (**d**) Cluster 5 with 14 items. Individual circle sizes indicate the relative frequency of occurrence of each keyword.

**Table 1 microorganisms-09-00419-t001:** Summary overview of main subject areas addressed in oil-related publications in the ScienceDirect database relating to the maritime Antarctic over the years 1970–2020.

Decade	Subject Areas	Example References
1970–1980	Distribution of hydrocarbons in the marine environment, pollution studies, toxicity assessments of Antarctic marine organisms (plankton, macrobenthos)	[20,28]
1981–1990	Chemical studies on alkane chains found in Antarctic marine sediments, chlorinated hydrocarbons	[21,29,30]
1991–2000	Toxicity studies on Antarctic fish, indigenous hydrocarbon-degrading bacteria, intertidal sediment studies, Bahia Paraiso, monitoring of hydrocarbon levels in seawater	[22,23,31]
2001–2010	Biodegradation studies by marine bacterial species, bioremediation, molecular and physiological studies of marine bacteria	[24,25,32,33]
2011–2020	Dynamic changes to microbial communities, genome sequencing, microcosm studies, biodegradation studies by bacterial consortia, hydrocarbon monitoring studies	[14,18,26,34,35,36,37,38,39,40,41,42]

**Table 2 microorganisms-09-00419-t002:** Antarctic studies related to hydrocarbon pollution published in the period 2000–2020.

Year of Publication	Location of Study	Hydrocarbon Studied	Reference
2003	Rod Bay, Ross Sea	Crude oil, diesel	[31]
2004	Terra Nova Bay Ross Sea	Diesel	[32]
2004	Admiralty Bay, King George Island	Diesel	[43]
2004	Terra Nova Bay, Ross Sea	Diesel	[25]
2007	Casey Station	Lubricant oil	[44]
2010	Victoria Land Coast	Diesel	[24]
2014	Davis Station	Diesel	[45]
2016	Rod Bay, Ross Sea	Diesel	[36]
2017	Carlini Station	Diesel	[14]
2010	Casey	Diesel, lubricant oil	[46]
2017	O’Brien Bay	Diesel, Lubricant oil	[12]
2020	Casey Station	Petroleum hydrocarbons	[26]
2016; 2017; 2020	Davis Station	Special Antarctic Blend (SAB) diesel, Marine Gas Oil (MGO) Lubricant oil, Intermediate fuel oil (IFO-180)	[37,40,47]
2020	McMurdo Station	Polyaromatic hydrocarbons (PAH)	[18]

**Table 3 microorganisms-09-00419-t003:** Studies confirming Antarctic marine bacterial ability to biodegrade various hydrocarbons.

Location	Organism/ Consortium	Hydrocarbon Source	Reference
Rod Bay, Ross Sea	*Oleispira antarctica* RB-8^T^	Crude oil, diesel	[31,36]
Marine sediment, Casey Station	Microbial community	SAB diesel, lubricant	[33]
Surface seawater, Victoria Land Coast, Ross Sea	Bacterial community	Diesel	[24]
Antarctic seawater	*Oceanobacillus* sp.	Lubricant oil, crude oil, diesel kerosene	[51]
Coastal sediment, King George Island	Bacterial community	Diesel	[2]
Antarctic marine sediments	Microcosm	Diesel, crude oil	[27]
Antarctic pristine seawater, Cape Legoupil	Bacterial community	Diesel	[50]

**Table 4 microorganisms-09-00419-t004:** The number of occurrences and total link strength of keywords pertaining to the subject of fuel and hydrocarbons in publications between 2000 and 2020 identified in the ScienceDirect database.

Keyword	Occurrences	Total Link Strength
Antarctica	444	1784
Southern Ocean	227	984
Pollution	140	663
Bioremediation	119	417
Biodegradation	109	376
Hydrocarbons	81	278
Antarctic	69	262
Bacteria	60	174
Polycyclic aromatic hydrocarbons	64	175
Petroleum hydrocarbons	42	156
Oil	31	136
Petroleum	30	137
Crude oil	22	132
Diesel	21	62
Fuel	16	85
Diesel oil	11	50
Fossil fuels	9	52
SAB	6	30
Special Antarctic Blend diesel	5	20
Diesel fuel	5	25
Lubricant oil	5	15
Oil spill	43	146
Fuel spills	9	36
Aromatic hydrocarbons	11	39

**Table 5 microorganisms-09-00419-t005:** Examples of studies using molecular approaches to investigate Antarctic marine microorganisms and their potential application in biotechnology.

Organism	Method of Study	Location	Application	References
Marine sediments, *Pseudomonas* and *Athrobacter* sp.	Cell culture	Fildes Peninsula	Antibiotic and drug resistance mechanisms	[128]
Macroalgae and fungal isolates	Cell culture	Potter Cove	Bioprospecting for antifungal activity	[129]
Macroalgae*: Psychroserpens* sp. NJDZ02	Whole genome sequencing	King George Island	Sodium alginate degradation, potential application in enzymatic industries	[130]
Marine microbiota	Metagenomics Illumina sequencing	Antarctic surface- and deep- seawater	Global ocean microbiota studies	[34]
Psychrophilic yeast - *Glaciozyma antarctica* PI12	Expressed Sequence Tags (EST)	Antarctic sea ice	Antifreeze proteins, expansin-like proteins	[126,131]
*Paenisporosarcina antarctica* CGMCC 1.6503^T^	Whole Genome sequencing	King George Island	Antifreeze proteins, Cold shock proteins, Osmotolerance	[126]
Sediment bacterial sample	T-RFLP coupled with DGGE	Livingston Island	Highlights the selection of microbial consortia with higher potential in response to oil spills in polar environments	[27]
Actinobacterial community	Geographic Information System (GIS)	Indian Ocean Sector of the Southern Ocean	Geospatial studies of Antarctic marine microbiome	[132]
Diatom: *Phaeodactylum tricornutum*	Cell immobilisation	Bellinghausen Dome, Ardley Cove.	Biomonitoring and evaluation of trace element bioavailability and potential transfer into marine food chains in Antarctica	[133]
Surface seawater bacterial community	Metagenomics: 454 pyrosequencing and MISEQ	Southern Ocean	Genes related to secondary metabolites of potential interest	[35]
Marine sediment, coastal seawater microbial samples	Laboratory setup	South Shetland Islands	Glycolipid surfactant, a potential biosurfactant	[42]
*Oleispira antarctica* RB 8	Whole Genome sequencing	Isolated fromAntarctic seawater	Osmoprotectants, alkane monooxygenase pathways, gene-transfer events	[125]
*Pseudoalteromonas, Psychrobacter, Arthrobacter* members.	Cell culture	Terra Nova Bay (Ross Sea)	Biodegradation of polychlorinated biphenyl	[134]
*Pseudoalteromonas haloplanktis* TAE 79	Cell culture	Antarctic seawater	Polar-active enzyme β-galactosidase, a potential candidate for lactose removal from dairy products at low temperatures	[135]

## Data Availability

No new data were created or analysed in this study.

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
