# Peer review of "Oil Bioremediation in the Marine Environment of Antarctica: A Review and Bibliometric Keyword Cluster Analysis"

_microorganisms, 2021, doi:10.3390/microorganisms9020419_

Round 1
Reviewer 1 Report
I consider that the paper, being a bibliometric review, has a high quality and includes the most interesting aspects about the information available to date on the use of microorganisms in the degradation of oils in Antarctica.
Author Response
Thank you for your comments.

Reviewer 2 Report
The Review is comprehensive and of value to those studying this area of research. While the bibliometrics is interesting, showing relationships within a subject area, the distinctiveness of colours and lines in Figures 6, 7, and 8 were diminished.
Is it possible to present this data in a more defined fashion, so that data is not lost in hardcopy printouts?
Author Response
Comment: The Review is comprehensive and of value to those studying this area of research. While the bibliometrics is interesting, showing relationships within a subject area, the distinctiveness of colours and lines in Figures 6, 7, and 8 were diminished.
Is it possible to present this data in a more defined fashion, so that data is not lost in hardcopy printouts?
Answer: Dear reviewer, unfortunately the pictures 6, 7 and 8 provided are software generated upon saving the picture. Figures are not screenshots but the actual image provided by the software.
